# Evaluation of Water Sorption and Solubility of 3D-Printed, CAD/CAM Milled, and PMMA Denture Base Materials Subjected to Artificial Aging

Mariya Dimitrova [1,*], Angelina Vlahova [1,2], Ilian Hristov [1], Rada Kazakova [1,2], Bozhana Chuchulska [1], Stoyan Kazakov [3], Marta Forte [4,*], Vanja Granberg [4], Giuseppe Barile [4,*], Saverio Capodiferro [4,†] and Massimo Corsalini [4,†]

1   Department of Prosthetic Dentistry, Faculty of Dental Medicine, Medical University—Plovdiv, 4000 Plovdiv, Bulgaria; angelina.vlahova@mu-plovdiv.bg (A.V.); ilian.hristov@mu-plovdiv.bg (I.H.); rada.kazakova@mu-plovdiv.bg (R.K.); bozhana.chuchulska@mu-plovdiv.bg (B.C.)

2   CAD/CAM Center of Dental Medicine, Research Institute, Medical University—Plovdiv, 4000 Plovdiv, Bulgaria

3   Oral Surgeon, Private Dental Practice—Sofia, 1000 Sofia, Bulgaria; kazakovstoyan@gmail.com

4   Department of Interdisciplinary Medicine, University of Bari Aldo Moro, 70100 Bari, Italy; vanjagranberg@libero.it (V.G.); saverio.capodiferro@uniba.it (S.C.); massimo.corsalini@uniba.it (M.C.)

*   Correspondence: maria.dimitrova@mu-plovdiv.bg (M.D.); fmarta@live.it (M.F.); giuseppe.barile@uniba.it (G.B.)

†   These authors contributed equally to the paper.

**Abstract:** Background: This in vitro study aimed to investigate and evaluate the values of water sorption and water solubility of four types of denture base polymers—3D-printed NextDent 3D Denture + (NextDent, 3D Systems, Soesterberg, The Netherlands), CAD/CAM milled Ivotion Base (Ivotion Denture System, Ivoclar Vivadent, Schaan, Liechtenstein), PMMA conventional Vertex BasiQ 20 (Vertex Dental, 3D Systems, Soesterberg, The Netherlands), and conventional heat-cured BMS (BMS Dental Srl, Rome, Italy)—which were subjected to artificial aging. Materials and methods: 200 specimens were created (*n* = 50), dried, and weighed accurately. They were immersed in artificial saliva (T1 = 7 days, T2 = 14 days, T3 = 1 month) and re-weighed after water absorption. After desiccation at 37 °C for 24 h and then at 23 ± 1 °C for 1 h, samples were weighed again. Next, thermocycling (100 h, 5000 cycles, 5–55 °C) was performed, and the water sorption and solubility were re-measured. IBM SPSS Statistics 0.26 was used for data analysis, revealing a direct correlation between water sorption and material type. Thermocycling at 55 °C increased water sorption for BMS and Vertex BasiQ 20. In conclusion, NextDent's 3D-printed resin had higher water sorption values throughout the study. Water solubility averages decreased over time, reaching the lowest in the 30-day period for CAD/CAM milled dental resin Ivotion Base. The artificial aging had no effect on Ivotion Base and NextDent's water sorption. Thermocycling did not affect the solubility of the materials tested. The conducted study acknowledges the great possibilities of dental resins for additive and subtractive manufacturing for the purposes of removable prosthetics in daily dental practice.

**Keywords:** 3D printing; digital dentures; water sorption; water solubility; dimensional stability; removable dentures

## 1. Introduction

There are numerous dental resins available today for the fabrication of removable dentures [1]. Polymethyl methacrylate (PMMA) is the most used dental acrylic and is utilized in the traditional heat polymerization procedure [2]. Polymethyl methacrylate holds numerous advantageous attributes, positioning it as the most pertinent material for the fabrication of removable prostheses [3–5]. These include easy processing with

uncomplicated techniques and equipment, favorable aesthetic qualities, high resistance to chemical degradation within the oral environment, and cost-effectiveness [6,7].

With the advancement of current technology, three-dimensional (3D) printing expands the options for creating removable dentures and saves a significant amount of time and effort for dentists and dental technicians [8–10]. The additive manufacturing process is based on stereolithography (SLA) and includes methods for fabricating items layer by layer [4]. The properties of the CAD/CAM materials differ because of the purpose they are dedicated to, and different physio-mechanical changes may occur during their use [11,12]. Compared to the traditional heat-polymerized PMMA used in removable dentures, 3D-printed dental resins offer several benefits. These encompass a streamlined laboratory process, time efficiency, and improved prosthetic-restoration planning [1,5,7]. However, both material types share drawbacks, such as eventual discoloration, which adversely impacts both the aesthetics and the lasting success of prosthetic treatments [3,10]. Several research studies have indicated that 3D-printed dental resins offer superior optical durability compared to PMMA resins in the context of removable prosthodontics [2,4,9]. The optical characteristics of denture base resins are highly significant, as they guarantee both aesthetic appearance and patient comfort [6].

For the manufacturing of complete dentures composed of polymethyl methacrylate (PMMA)-based resins, various processes, such as dough molding and compression or injection molding, have been used [13]. On the other hand, traditional approaches incorporate several laboratory procedures [14]. As digital technology has advanced, new computer-aided design and computer-aided manufacturing (CAD/CAM)-based techniques for material processing in dentistry, such as subtractive milling (SM) and additive manufacturing (AM), have arisen [15–17]. Even though both approaches rely on a digital 3D model created by CAD software for denture manufacturing, the two methods differ significantly. In the milling technique, a complete denture is crafted at a milling station using a pre-polymerized polymethylmethacrylate block produced with high pressure. A significant drawback arises from the substantial product wastage following the completion of the milling procedure [2]. On the other hand, additive manufacturing (3D printing) represents a digital technique, in which 3D objects are created by successively depositing material in layers to create a model [18–20]. Complete denture resins fabricated using CAD/CAM milling and rapid prototyping exhibit comparable levels of biocompatibility and surface roughness [21,22]. Nonetheless, the milled denture resins outperform the rapidly prototyped ones in terms of their mechanical characteristics. The arrangement of printing and the specific 3D printer utilized can impact both the strength of the resin and the texture of the surface [23,24].

Technologies such as additive manufacturing and computer-aided design (CAD) are having a substantial impact on all parts of dentistry [2]. Three-dimensional printing enables the accurate construction of one-of-a-kind, complicated geometrical forms in a range of materials and on a local or industrial scale from digital data [21]. Emerging AM technology is changing the clinical and laboratory processes for making removable dentures [4,7]. Even while practically everything we manufacture for our patients can now be made with a 3D printer, no single technology can meet all of their needs [3,23].

The subtractive or CAD/CAM milling procedure involves milling the CD from a commercially made pre-polymerized PMMA disc [9,10]. Many studies have shown that milled resins have superior mechanical and surface properties, comparable color stability, reduced microbial colonization [5,6], and a lower leech rate of residual monomer when compared to compression molding resins because they are manufactured under high-pressure and well-controlled conditions [2,25]. The additive manufacturing technique, also known as rapid prototyping or 3D printing, also comprises serial apposition of the liquid resin material on a support structure followed by curing with visible light, ultraviolet light, heat, or a laser [3]. This layering and curing process is repeated until the CD shape specified in the CAD is accomplished [7].

Water sorption is the ability of dental materials to absorb liquids and change their volume and weight [26,27]. This is a physical and chemical process, which can be defined as a phenomenon of fixation or capture of a gas or a vapor (sorbate) by a substance in a condensed state (solid or liquid) called sorbent [5]. Water solubility is a measure of the amount of chemical substance that can dissolve in water at a specific temperature. Solubility is generally expressed as the number of grams of solute in one liter of saturated solution [7,18,22]. Denture base resins have low water solubility, which results from the leaching out of unreacted monomer and soluble additives into the oral cavity. This is an undesired property and may cause soft-tissue reactions [4].

Dental resins must not change their dimensions over time and must be resistant to volumetric changes under all conditions [28,29]. The fitting of the denture base to the alveolar ridges is significant for the retention of the removable denture [8]. Water sorption properties inevitably affect the volumetric changes in dental resins, which might lead to the aging of the material and, moreover, might seriously affect the stability of the denture during masticatory function [30–32]. On the other hand, water sorption increases the dimensions of the dentures and compensates for shrinkage, which occurs during the polymerization process, and affects both conventional and 3D-printing techniques [9,33].

Conventional denture base resin has better flexural strength compared to 3D printed, which shows inferior surface roughness and lower hardness values than the heat-cured materials for removable dentures [8,12,27].

As the polymer material dries, the water is eliminated, and the polymer chains return to their original position. If rewetting follows, the polymer chains expand again [10]. In this way, a cycle of micro-movements of the chains is created, and microcracks appear between the individual macromolecules, which can lead to fracture of the removable denture after mechanical loading [11,12]. Saliva absorbed for one month leads to a linear expansion of 0.03%, and after nine months, it expands by 0.04 % [13]. A decrease in moisture or the amount of solvent in the atmosphere results in further drying of the materials [33,34]. This process continues until a certain percentage of drying of the substance is achieved or until the desiccant is exhausted [14]. In addition to removing moisture from substances, solvents can also be removed, depending on the choice of the appropriate desiccant [15,21].

According to the study by Perea-Lowery et al. [35], the water sorption of 3D-printed plastics has a higher value compared to heat-polymerized ones. This could be related to the polymerization process of conventional polymers, which takes place at a higher temperature and for a longer period. Thus, it causes reduced water sorption, water solubility, and residual monomer concentration, which have been demonstrated in several studies [36].

In addition, differences in the chemical composition of 3D-printed and PMMA acrylics for removable dentures must be considered, as the type of dental resin plays a significant role in the level of water absorption and water solubility [37–39]. Photosensitive thermoset liquid monomers such as urethane dimethacrylate (UDMA) and triethylene glycol dimethacrylate (TEGDMA), photo-initiators, and additives are used for the manufacture of printable resins [7,14,15]. A free-radical polymerization reaction begins when those monomers are exposed to a sufficient light source. Terminal aliphatic C=C connections are broken and transformed to main C-C covalent bonds between methacrylate monomers during this process, causing the material to transition from a fluid to a solid state [19,40].

A very interesting aspect that occurred in the literature recently was the fractal dimension and texture analysis of materials [41,42], which showed that the flexural strength of the tested samples had lower values [43].

Thermocycling is a laboratory technique for exposing dental materials to temperature ranges similar to those found in the oral cavity, which can have negative consequences due to different coefficients of thermal expansion between the mucosa and the tested materials [12,44]. Thermal cycling is considered to be one of the most severe thermal environments.

The purpose of the current study was to evaluate the water sorption and the water solubility of two types of denture base resins—3D-printed dental resin and PMMA heat-

polymerized conventional-type acrylic. In order to assess the influence of treatment and storage conditions on the outcome variables, the following hypotheses were tested:

**H$_0$.** *Water sorption/water solubility and thermocycling will not significantly influence the dimensions and the mass of the two groups of test samples.*

**H$_1$.** *Water sorption/water solubility and thermocycling will significantly influence the dimensions and the mass of the two groups of test samples.*

## 2. Materials and Methods

For the aim of the current study, 200 ($n = 50$) samples, divided into four groups, were prepared in the shape of a parallelepiped with dimensions of 20 mm by 20 mm in width and length, respectively, and 3 mm in cross-sectional diameter. The shape of the test specimens was modified exclusively for our research and following many similar studies [24–27]. The methodological protocol was in accordance with the ISO standard (ISO 20795 1: 2013) [19]. The shape and size of the test specimens were designed according to the predetermined criteria using the non-parametric software Free CAD Version 0.19 (Hanau, Hessen, Germany), and an STL file was created for this purpose.

The first and second groups of experimental samples were made by Vertex BasiQ (Vertex Dental, 3D Systems, Soesterberg, The Netherlands) and BMS 014 (BMS Dental Srl, Rome, Italy) with two types of heat-polymerizing acrylic using a conventional flasking method (Table 1).

**Table 1.** Denture base acrylic resins used in the study.

| Product | Composition | Detail | Manufacturer |
|---|---|---|---|
| BMS 014 | Methyl methacrylate | Heat curing | BMS Dental Srl, Rome, Italy |
| Vertex BasiQ 20 | PMMA | Heat curing | Vertex Dental, 3D Systems, Soesterberg, The Netherlands |
| NextDent Denture 3D+ | Methacrylate-based photopolymerized resin | 3D printing | NextDent, 3D Systems, Soesterberg, The Netherlands |
| Ivotion Base | Polymethyl methacrylate, co-polymers | CAD/CAM milling | Ivotion Denture System, Ivoclar Vivadent, Schaan, Liechtenstein |

The wax samples were shaped and sized to match the test body accurately. Then they were embedded in a dental stone (Elite Model, Class III, Zhermack, Rovigo, Italy) within a metal flask, following the guidelines provided by the manufacturer. Once the dental stone had solidified, the flask containing it was immersed in boiling water for a span of 3 min. Subsequently, the flask was taken out of the water and allowed to sit undisturbed for an additional 3 min. Following this, the flask was opened, and the wax was removed. This was accomplished by utilizing a blend of warm water and detergent, succeeded by a round of boiling water, followed by a cooling phase. Once the cooling was completed, a separation agent was administered to the surface of the plaster. For optimal results, the manufacturer recommended the use of BMS ISOLANT (Spofadental, Jičín, Czechia). The polymer was pre-weighed and mixed with the monomer in a weight ratio of 2:1 to create the experimental bodies of thermosetting plastic. The polymer and monomer were combined at room temperature in a clean porcelain vessel according to the manufacturer's specifications before being pressed for 30 min using a pneumatic press (Dentalfarm, Srl, Torino, Italy). The acrylic was then polymerized using a heat-curing apparatus (KaVo Elektrotechnisches Werk, Leutkirch, Germany). The temperature of the water gradually increased from room temperature to 74 °C for 8 h, and then to 100 °C for 0.5 h. Flasks were allowed to cool after polymerization and before specimen recovery. They were then finished with tungsten carbide burs to remove the flashes (HM79GX-040-HP, Meisinger, Centennial, CO, USA) at 18,000 rpm, and all of the surfaces were polished with silicon

carbide papers with gradually increasing grits, beginning with 320, then 400, and finally 600, and slurry pumice with a soft bristle brush were used for the final polish (Steribim Super, Bego, Wilhelm-Herbst-Strabe, Germany). The third group of experimental bodies was made using the 3D printing method of NextDent Denture 3D+ (NextDent, 3D Systems, Soesterberg, The Netherlands) for dental resin for removable dentures. The test samples were 3D printed with dental resin for denture bases using a pre-generated STL file from the software package. The STL file format, which replaces an object's geometry with triangles, is a standard for communicating three-dimensional information to 3D printers. NextDent's 3D printer (NextDent 5100 DLP 3D Printer, 3D Systems, Budel, the Netherlands) was used to apply the technology. Dental resin is a photopolymerizable liquid material that is based on polymethyl methacrylate. It was inserted into the printer's bottom tray.

The process in the 3D printer software could begin after the file was imported and the 3D pictures of the trial bodies were positioned. The resin specimens were printed with a layer thickness of 50 μm. This method required 95 min of exposure time. The final experimental bodies were placed on a special platform at the top of the printer after printing. Additional processing was performed on the samples, including the removal of unpolymerized material from the surface by immersing them in isopropyl alcohol for 10 min. Isopropyl alcohol (IPA-2-propanol or rubbing alcohol) is a clear, powerful cleaning medium that can be used on a variety of 3D printing materials.

The method for 3D-printed elements normally takes six minutes, and the IPA is dissolved in distilled water at a 70% isopropyl alcohol: 30% distilled water ratio. After washing, the test pieces were immersed in glycerin for 45 min in the final polymerization furnace to react with the remaining monomers.

The last group of experimental samples was made from CAD/CAM resin discs for subtractive manufacturing Ivotion Base (Ivotion Denture System, Ivoclar Vivadent, Liechtenstein). A wax prototype of the specimen was scanned, and the resultant scan data were stored in standard tessellation language (STL). The data were transferred to CAM software V4, Version 4.0 (PrograMill CAM software). Then a computer-aided machine automatically milled the designed specimens from pre-polymerized CAD/CAM resin discs (100% weight, PMMA) (Ivotion, Ivotion System, Ivoclar Vivadent, Liechtenstein) using a subtractive technique. During the milling process, burs with a maximum diameter of 2.5 mm and 5-axis were utilized to accurately produce more fine details and to avoid overheating in wet conditions, according to the manufacturer.

After the test samples were dried to their optimal mass, they were weighed using analytical balances (Mettler—Toledo, Milan, Italy) with an accuracy of 0.0001 g to obtain optimally precise results.

This was followed by immersion in artificial saliva, prepared by a chemist according to a specific formula, for three periods (T1 = 7 days, T2 = 14 days, T3 = 1 month) and calculating the new mass of the test bodies after water absorption (Table 2).

**Table 2.** Chemical composition of artificial saliva used in this study.

| Component | % |
|---|---|
| NaCl | 0.0856 |
| KCl | 0.1200 |
| $MgCl_2 \times 6H_2O$ | 0.0052 |
| Mannitol | 0.2000 |
| $K_2HPO_4$ 0.0456 | 0.0456 |
| Carbomer 974P 0.1000 | 0.1000 |
| NaOH 10% 0.4000 | 0.4000 |

**Table 2.** *Cont.*

| Component | % |
|---|---|
| $CaCl_2 \times 2H_2O$ 0.0148 | 0.0148 |
| $KH_2PO_4$ 0.0272 | 0.0272 |
| Purified Water 96.9016 | 96.9016 |

Figure 1 represents the planned shape of the 3D model, which recreates the actual geometry of the test bodies.

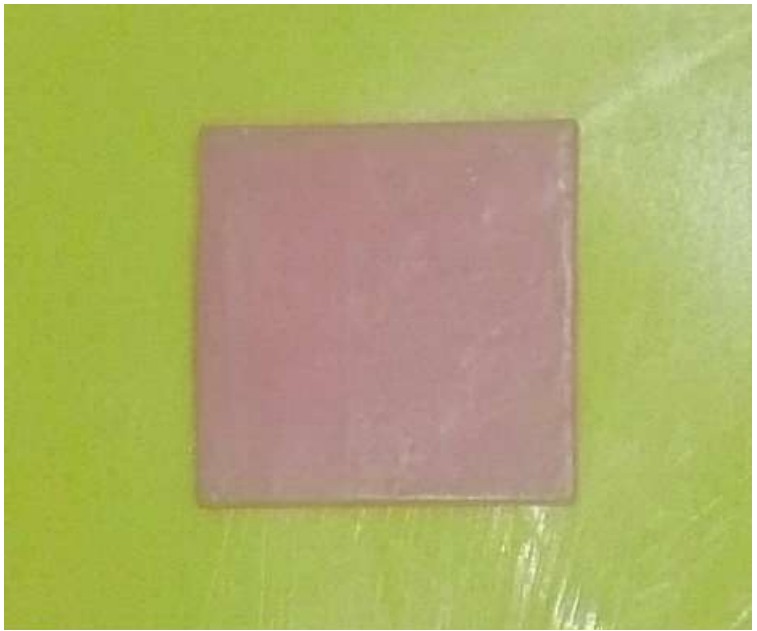

**Figure 1.** A 3D model of the dimensions of the resin specimens using the software 3D-Viewer.

The test samples were placed on a porcelain tile in one of the desiccators. The desiccator (ISKO, Gupta Scientific & Glass Works, Haryana, India) was stored at a temperature of 37 °C for 24 h. Then the test samples were transferred directly into the second desiccator, into which pre-dried silica gel was placed. The temperature conditions in which the second desiccator was stored for 1 h were 23 ± 1 °C. In order not to disturb the vacuum environment, it was of utmost importance that the desiccator was tightly sealed during the test, except for the period necessary to remove and replace the test samples.

The same work cycle was continued until a consistent mass, known as the conditioned mass, was obtained, i.e., when the mass loss of each sample between subsequent weighing was less than 0.2 mg. The volume (V) of each specimen was calculated at this point by taking the average of three length measurements and the average of five thickness measurements. The specimen's thickness was measured in the middle and at four evenly spaced places 1.5 mm from its edge. During the measurements, the temperature storage conditions were at room temperature. The volume of each test specimen was defined by the mathematical formula of a rectangle (Equation (1)):

$$V = L \times W \times H \tag{1}$$

Equation (1). The volume of a rectangular equation, where:

V—volume;
L—length;
W—width;
H—height (thickness).

### 2.1. Water Sorption

The water sorption value for each sample was expressed in micrograms per cubic millimeter using the following equation:

$$\text{Wsp} = \frac{m2 - m3}{V}$$

(2)

Equation (2). Water sorption equation, where:

Wsp—water sorption;
m2—the mass of the test body after immersion in artificial saliva;
m3—the mass of the reconditioned experimental body in micrograms;
V—the volume of the test sample.

The samples were submerged in artificial saliva for 7 days at 37 °C. Following this period, polymer-coated tweezers were used to extract the samples from the solution and to dry them with a clean, dry cloth until no visible moisture remained. They were then weighed for 60 s after being removed from the artificial saliva, and the mass was calculated as m2. This was the weight of the test body after liquid absorption. Similarly, the same methods were applied for the remaining two periods of the study—14 days and 1 month (Figure 2).

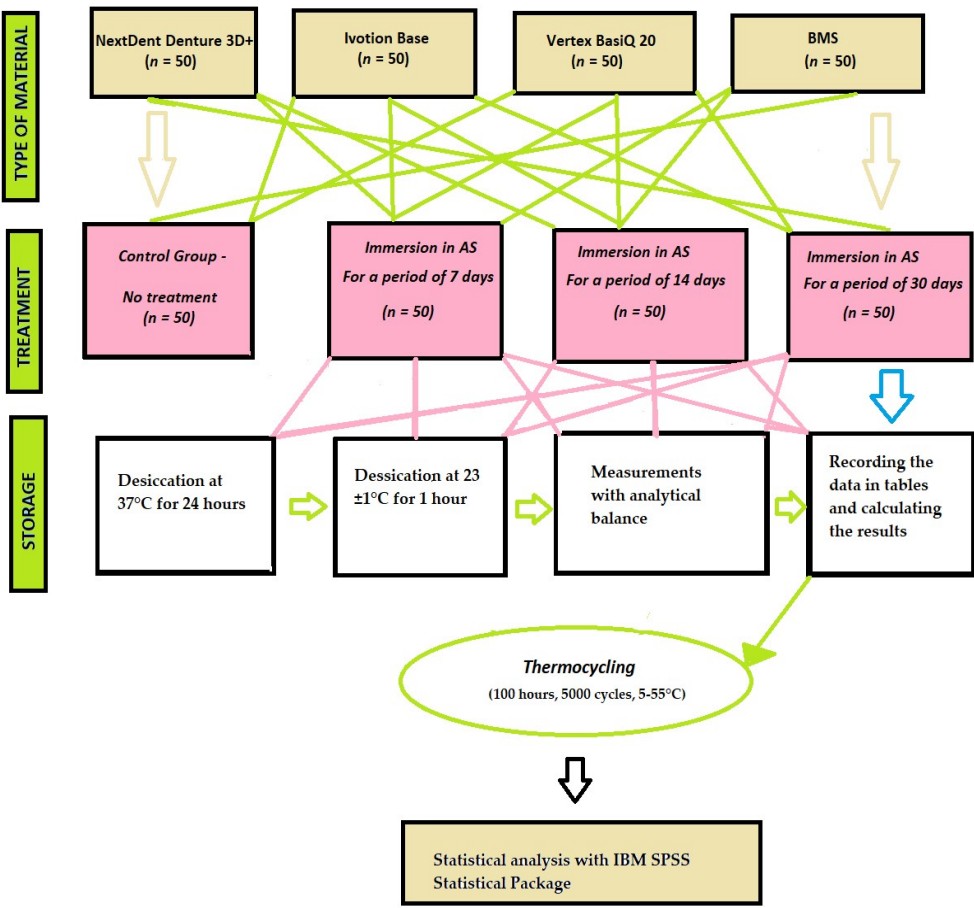

**Figure 2.** Design of the study.

After weight recovery, the test samples were conditioned to a constant mass in the desiccator under the conditions already described. The masses of the reconditioned samples were referred to as m3. To obtain objective results, it was essential to apply the same conditions as in the first drying process (temperature, time, etc.), using the same number of test specimens and freshly dried silica gel.

### 2.2. Water Solubility

The experimental setup from the first subtask was used to complete this task. The test specimens were placed in a desiccator filled with dried silica gel, which was utilized to keep the plastic specimens at 37 °C (1 °C) for 24 h. After that, all test subjects were placed in a second desiccator for 1 h at 23 °C (1 °C). They were then weighed to the nearest 0.1 mg with an analytical balance. The drying cycle was repeated indefinitely until a consistent mass (m1) was reached. The dried test specimens were immersed for three different periods of time—7 days, 14 days, and 1 month—at a temperature of 37 °C (±1 °C). After removing the plastic samples from the containers, they were left outside for 60 min and dried with a clean paper towel before being weighed again (m2).

The water solubility values of the test bodies were calculated using the ISO standard formula. Data were statistically processed using the IBM SPSS Statistics Version 26 software program (SPSS Inc., Chicago, IL, USA).

Water solubility was determined by the ratio of solute per unit volume exposed during immersion and expressed in micrograms per cubic millimeter for each test body using the following formula (Equation (3)):

$$Wsl = \frac{m1 - m3}{V} \tag{3}$$

Equation (3). Water solubility equation, where:

Wsl—water solubility;
m1—the conditioned mass of the sample;
m3—the mass of the reconditioned experimental body;
V—the volume of the experimental body;

Means were statistically treated with one-way analysis of variance (ANOVA) followed by Duncan's multiple range test to determine the significant difference between groups at a $p < 0.05$ level of significance.

### 2.3. Thermocycling

Artificial aging was applied to the tested groups using the thermocycling device LTC 100 (LAM Technologies, Italy). The LTC100 thermocycler is designed to simulate the temperature variations that dental materials undergo in the oral cavity. The apparatus is completely automatic and capable of complex and continuous simulations. Each group's samples were kept in distilled water (37 °C). Thermocycling was performed with 5000 cycles between 5 °C and 55 °C (with a dwell time of 30 s). The simulation cycle was built as a sequence of one or more profiles; each one could be repeated once or more. A profile defines all the conditions in which the thermal changes occur. The automatic tank covering system eliminates liquid evaporation while working, and the liquid level in the tanks is automatically maintained constant through liquid level sensors and the reservoir tank. This procedure is equivalent to a five-year cycle of oral temperature conditions [10]. The obtained data were statistically analyzed (using one-way ANOVA), and the mean values were compared using the Tukey test ($\alpha = 0.05$).

## 3. Results

### 3.1. Water Sorption

To statistically process the data obtained from the study, a statistical method for dispersion analysis was applied (one-way ANOVA analysis) for the three periods (with a confidence interval of $\alpha < 0.05$), with an aim to find out if there was a correlation between the type of material and the measured values for water sorption (Figure 3).

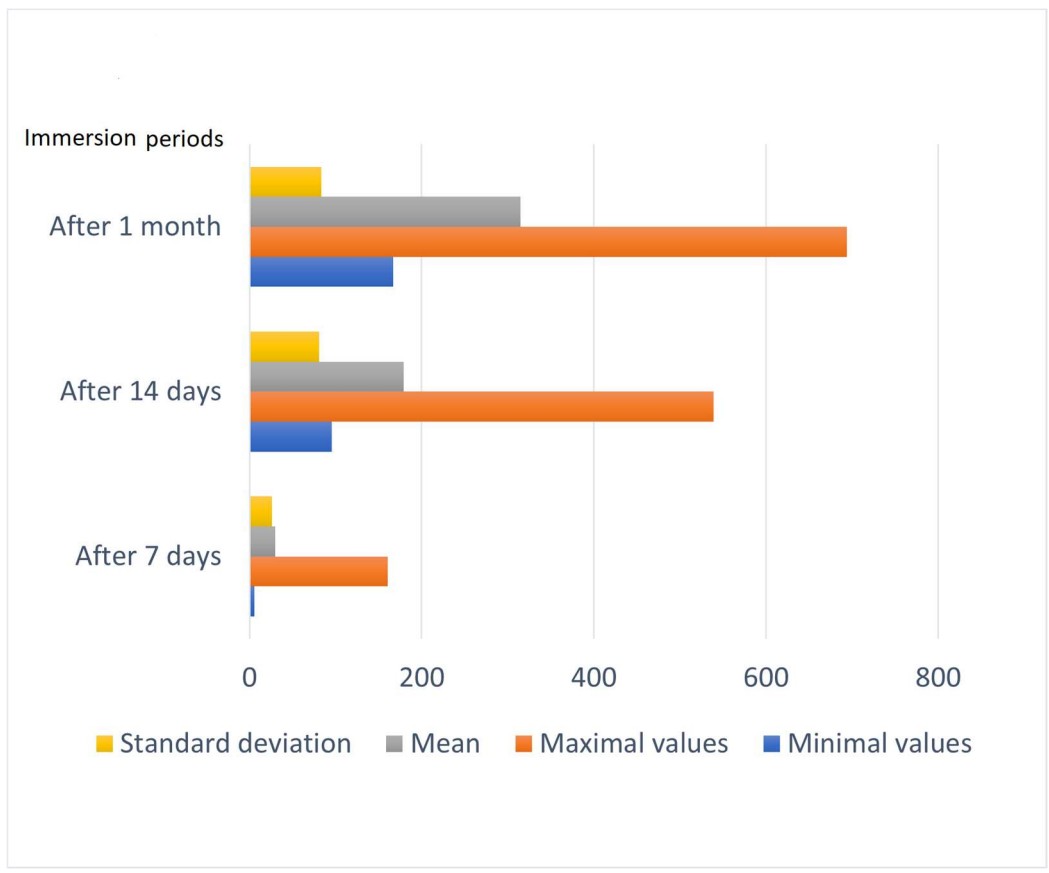

**Figure 3.** Water sorption values of the tested samples for different periods.

During a test comparing the obtained values for water sorption after 7 days with the type of material, it was found that the obtained value for $p$ was a much smaller value than $\alpha < 0.05$. As a result, there was a direct relationship between the material type and the water sorption values produced.

When the statistical method was carried out analogously for a period of 14 days, during which the obtained values for water sorption were compared with the type of material, it was discovered that the obtained value for $p$ was substantially less than 0.05. Therefore, we can conclude that there was a direct correlation between the type of material and the obtained values for water sorption for a period of 14 days (Table 3).

**Table 3.** One-way ANOVA—Immersion in artificial saliva for a period of 14 days.

| One-Way ANOVA Analysis | Sum of Squares | Degrees of Freedom | Mean of Squares |
|---|---|---|---|
| In between the groups | 249,217.7 | 1 | 249,217.7 |
| In the groups | 131,748.8 | 68 | 1937.482 |
| Total | 380,966.5 | 69 | |

In a test comparing the obtained values for sorption after 1 month with the type of material, it was found that the obtained value for $p$ was much less than $\alpha < 0.05$. Therefore, there was a direct relationship between the type of material and the sorption values obtained (Table 4).

**Table 4.** One-way ANOVA—Immersion in artificial saliva for a period of 1 month.

| One-Way ANOVA Analysis | Sum of Squares | Degrees of Freedom | Mean of Squares |
|---|---|---|---|
| In between groups | 479,605.2 | 1 | 479,605.2 |
| In the groups | 509,813.8 | 68 | 7497.262 |
| Total | 989,419.1 | 69 | |

The mean values for both studied materials were similar for the first 7 days of immersion in artificial saliva, as shown in Diagram 5, with the heat-polymerized material exhibiting greater changes. In both dental resins, there was an increase. For Ivotion Base and NextDent, the mean values for water sorption were the highest after 1 month of immersion in artificial saliva (Figure 4).

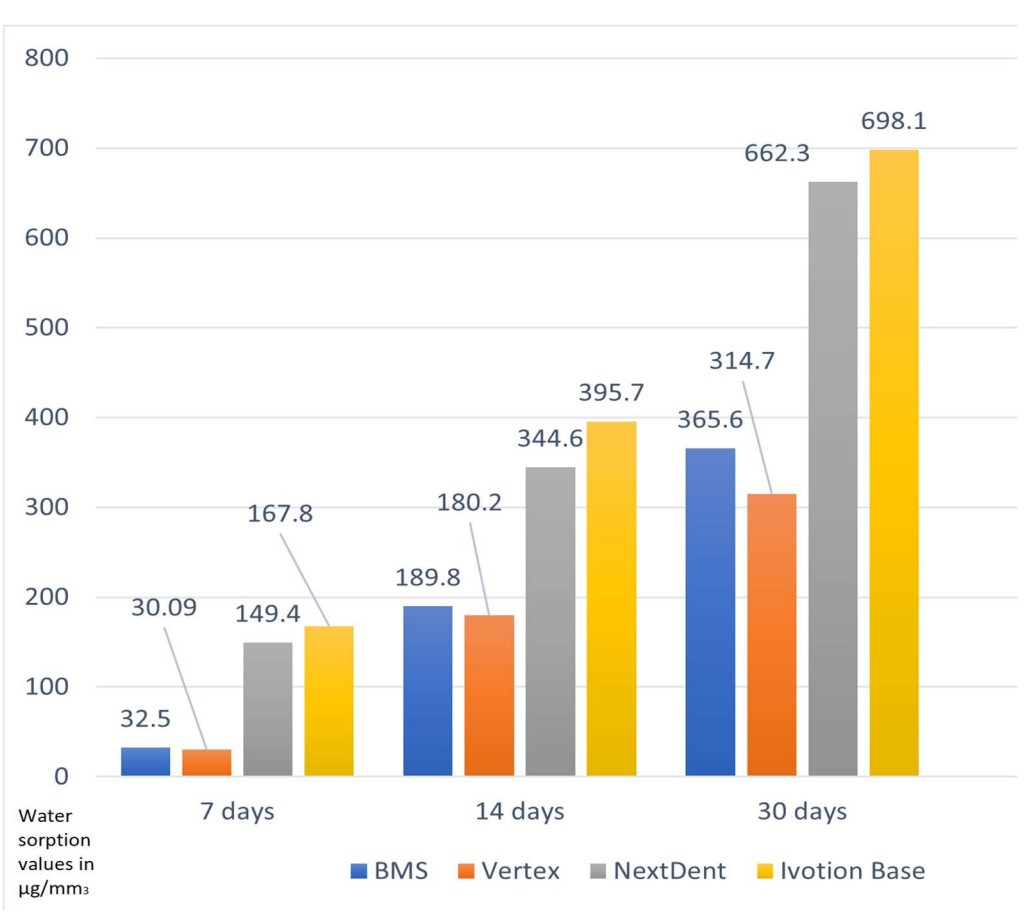

**Figure 4.** Mean values of the water sorption of the four dental resin types in the different observation periods.

The values for the water sorption of Vertex BasiQ 20 were more consistent during the three periods of time, and there was a steady increase, with the highest mean after 1 month. The highest values for water sorption after 30 days were for Ivotion Base, followed by NextDent.

### 3.2. Water Solubility

To statistically process the data obtained from the conducted research, a one-way ANOVA analysis was performed. To determine whether the obtained water solubility values had a correlational dependence on the type of material, we chose a confidence interval of $\alpha < 0.05$. The values for standard deviation and mean difference for all four

experimental groups are presented in Figure 5. Decreased water solubility is indicated as a negative numerical value.

**Figure 5.** Standard deviation values of Vertex BasiQ 20 for water solubility.

In the test conducted comparing the obtained values for water solubility after a period of 7 days with the type of material, we found that the value for *p* was a much larger value than $\alpha < 0.05$. Therefore, we can conclude that there was no direct correlation between the type of material and the obtained water solubility values (Table 5).

**Table 5.** One-way ANOVA for immersion period of 7 days.

| One-Way ANOVA Analysis | Sum of Squares | Degrees of Freedom | Mean of Squares |
|---|---|---|---|
| In between the groups | 6051.2 | 1 | 6051.2 |
| In the groups | 17,257.58 | 68 | 253.788 |
| Total | 23,308.78 | 69 | |

As a result of the statistical test conducted, in which the obtained values for water solubility after 14 days were compared with the type of material, it was found that the obtained value for *p* was a smaller value than $\alpha < 0.05$. Therefore, there was a direct correlation between the type of material and the obtained water solubility values (Table 6).

**Table 6.** One-way ANOVA—results after 14 days.

| One-Way ANOVA Analysis | Sum of Squares | Degrees of Freedom | Mean of Squares |
|---|---|---|---|
| In between the groups | 0.167 | 1 | 0.167 |
| In the groups | 574.61 | 68 | 8.45 |
| Total | 574.777 | 69 | |

For a period of 1 month after the test comparing the obtained values for water solubility with the type of material, it was proved that the obtained value for *p* was equal to 0.002, which was a smaller value than $\alpha < 0.05$. Therefore, we conclude that there was a direct correlation between the type of material and the water solubility values obtained (Table 7).

**Table 7.** One-way ANOVA for immersion period of 1 month.

| One-Way ANOVA Analysis | Sum of Squares | Degrees of Freedom | Mean of Squares |
|---|---|---|---|
| In between the groups | 8340.2 | 1 | 8340.2 |
| In the groups | 29,357.58 | 68 | 418.968 |
| Total | 40,508.78 | 69 | |

As a result of the processed statistical data, we can summarize that for a period of 7 days, there was no correlation between the period of stay of the materials in artificial saliva and the type of material itself. For the remaining periods, we observe that there was a dependence between the material type and the residence time of the materials in artificial saliva and, accordingly, the degree of their water solubility.

A correlational dependence was established between the studied period and the value of *p*, which progressively decreased with an increase in the stay of the materials in artificial saliva.

### 3.3. Thermocycling

A Tukey test was applied in order to compare each type of material before and after thermocycling (Figure 6). All *p* values were equal to 0.05, confirming that the process of artificial aging was significant and a factor in the property changes in the test specimens. For the three immersion periods after thermocycling, the values of the water sorption increased gradually in Vertex BasiQ 20 and BMS. The process of artificial aging did not affect the values of NextDent and Ivotion Base. Thermocycling did not influence the tested groups' water solubility values.

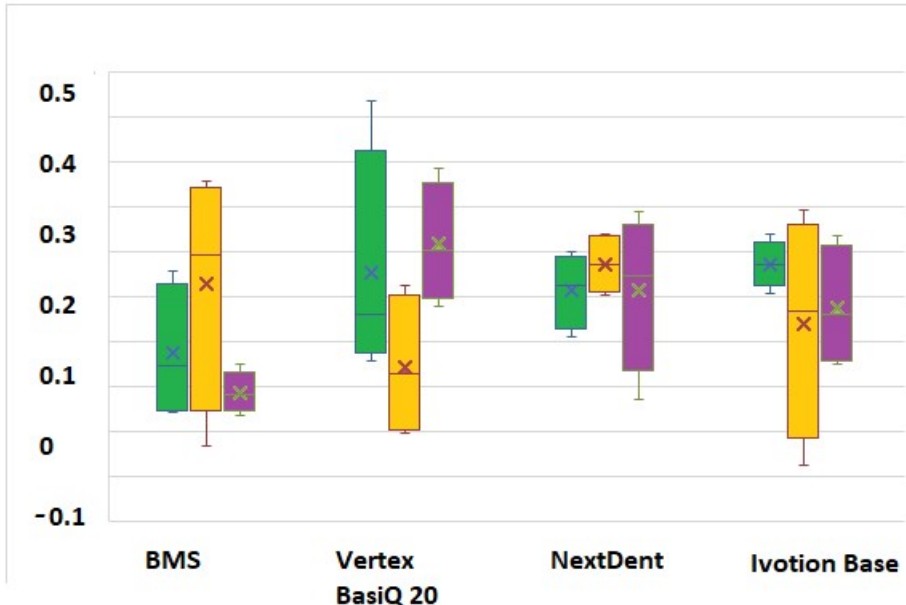

**Figure 6.** Tukey test for multiple comparisons: green color—immersion period of 7 days; yellow color—immersion period of 14 days; purple color—immersion period of 1 month.

## 4. Discussion

The findings show that both types of dental resin materials for removable dentures meet the requirements regarding water sorption and water solubility given in the standard ISO 20795-1, measured before and after thermocycling according to a modified method [45].

The study of Gad et al. revealed that the water sorption of 3D-printed resins increased in comparison to heat-polymerized resin [46]. This finding is similar to a previous study reporting that the sorption of 3D-printed acrylic resins increased in comparison to pressed resin for occlusal devices [47].

According to ISO 20795, during the storage of the specimens, 32 μg/mm$^3$ (water sorption) is the acceptable volumetric mass increase in denture base materials per volume [45]. All water sorption values of the tested 3D-printed specimens were lower than the ISO recommendation for maximum water sorption [48]. The results for the 3D-printed resin showed that they are suitable for clinical application [49]. However, further investigations are required to present the variations in water sorption between 3D-printed materials and conventional dental resins for removable dentures.

The findings of Vallittu et al. showed that the increased amount of water sorption could be caused by the printed layering technique [50]. The absorbed water enters between the layers into the resin's polymer; thus, the interpolymeric spaces are filled with water, which penetrates the empty spaces and voids, forcing the polymer chain away from other chains [51–54]. This theory was confirmed using the method of electron microscopy [55]. Water–polymer chain interactions may cause changes in mechanical strength, minor chemical degradation, and elution of residual monomers [56–58].

The results of the current study showed that the water sorption of 3D-printed resins increased in comparison to the conventional resin for removable dentures [59,60]. Water sorption is a diffusion-controlled process, which occurs through penetration into empty spaces, such as micro-voids, or by a molecular interaction [61,62]. The degree of resin polarity is of great importance for the progress of this process. As a result, water sorption is one of the most remarkable properties when assessing denture base durability [63,64].

In our study, two groups of test samples of denture base polymers were investigated, similar to the study of Bagheri et al. [65]. According to their investigation, excessive solubility of plastics for removable prosthetic structures can lead to surface deformations. While the solubility of methacrylate-based polymers in various solutions has been extensively studied, to our knowledge, very little information is available regarding the solubility of 3D-printed plastics for removable prosthetics [66,67].

In the current research, the experimental specimens were subjected to different storage conditions. The temperature in the first desiccator was 37 °C, while in the second, it was decreased to 23 ± 1 °C. Then all test samples were measured at room temperature. According to the investigation of Silva et al. [68], all these changes could influence many of the materials' mechanical properties, such as fracture toughness and the hardness of the polymers.

The average solubility values presented by the tested composite resins according to ISO 4049: 2019 ranged from 2.3 to 4.2 μg/mm$^3$; these values are lower than the maximum value established by the ISO 4049: 2019 standard (<7.5 μg/mm$^3$) [69]. The results of the current study showed that the solubilities of all the materials in all the solutions were acceptable according to ISO 4049: 2019. This coincides with the data obtained from our study on the solubility of the two types of plastics for removable prosthetic restorations.

According to the study by Labban et al. [70], the density in methacrylate-based resin composites can vary because of free-radical polymerization, causing heterogeneity in the polymerized material that can facilitate the trapping of residual monomers from where they can easily be removed. Compared to the polymerization of methacrylate-based plastics, the photoactivated cationic polymerization process of silorane resins is relatively insensitive to oxygen. This not only reduces polymerization shrinkage but also increases the degree of conversion [71,72]. In their study, Petropoulou et al. [46] compared the water solubility of four groups of experimental samples. As a result of the results obtained, it

was concluded that the reported solubility values were mainly influenced by the material characteristics, and the variations occurring between materials of the same type were attributed to differences in the composition of the polymer chains [73].

In our research, the test samples were subjected to artificial aging using thermocycling, which provided interesting results—the water solubility of Vertex BasiQ 20 increased significantly, while NextDent was not affected at all. Various aging approaches, including water storage, thermocycling, and storage in sodium hypochlorite (NaOCl) solution, either separately or in combination, have been utilized in other studies to predict the long-term clinical performance of adhesive systems [74,75]. Water storage aging involves submerging samples in distilled water at 37 °C for 3 to 12 months to partially mimic oral environmental conditions [23]. Meanwhile, thermocycling employs 5000 to 30,000 cycles of alternating hot and cold water to simulate the stresses of the oral environment, potentially leading to gap formation along the adhesive interface and allowing fluid infiltration. The storage method of immersing samples in a 10% NaOCl solution, lasting 1 to 5 h, leverages the proteolytic properties of hypochlorite to imitate aging effects by degrading the organic resin and tooth interface components, including unprotected collagen fibrils [76]. Three-dimensional printing materials, such as composite cements, are made of dimethacrylates or higher-functional materials (3–6 methacylate or acrylate groups), which means they are highly densely cross-linked, resulting in inferior sorption and solubility [77]. According to the study of Ghavami-Lahiji [78], the process of thermocycling significantly decreased the hardness and flexural strength of the experimental bodies. Furthermore, in the study of Yap et al. [79], for all treatment groups, the water solubility values of the experimental samples were not affected.

Thermocycling emerged as the technique that led to the most significant deterioration of the bonding interface and resulted in the weakest bonding strength [80]. After assessment, it was identified as the most suitable technique for simulated aging. As a result, it was chosen for the present study's objectives. Conversely, the storage method involving NaOCl demonstrated an enhancement in bond strength under the assessed circumstances.

The limitations of the conducted study can be summarized in the following:

- Because this experimental research provides such a high level of control, it can produce results that are specific and relevant with consistency. It is possible to determine the values of the water sorption and water solubility, making it possible to evaluate the properties of the two types of dental resin in a much shorter amount of time compared to other verification methods [55].
- Secondly, the data can be corrupted to seem like they are positive, but because the clinical environment is so different from the controlled laboratory environment, positive results could never be achieved outside of experimental research.

## 5. Conclusions

From the obtained results, it can be summarized and concluded that:

- The direct correlational dependence is obvious not only between the acquired values for water sorption but also between the kinds of materials, which is significant for the materials' dimensional qualities.
- For all four types of test samples, increasing the imbibition times increased the imbibition of artificial saliva.
- For all durations of the current study, the mean difference values for water sorption were higher for NextDent and CAD/CAM milled Ivotion Base 3D-printed resin for removable dentures. The average values for water solubility gradually decreased with time, reaching their lowest point after one month (30 days) for test samples manufactured with 3D-printed dental resin.
- Thermocycling of BMS and Vertex BasiQ 20 at an upper temperature of 55 °C enhanced water sorption significantly but did not influence NextDent readings.
- Thermocycling did not affect the water solubility of the materials studied.

**Author Contributions:** Conceptualization, M.D.; methodology, M.D. and A.V.; software, I.H., A.V. and V.G.; validation, I.H., R.K., G.B. and S.C.; formal analysis, B.C. and S.K.; investigation, M.D. and S.K.; resources, M.D. and S.K.; data curation, B.C., S.K., M.F. and V.G.; writing—original draft preparation, M.D.; writing—review and editing, A.V. and I.H.; visualization, M.D. and G.B.; supervision, B.C., R.K. and M.C.; project administration, R.K.; funding acquisition, R.K. All authors have read and agreed to the published version of the manuscript.

**Funding:** This research received no external funding.

**Institutional Review Board Statement:** Not applicable.

**Informed Consent Statement:** Not applicable.

**Data Availability Statement:** Not applicable.

**Acknowledgments:** We are grateful for the support of Medical University of Plovdiv and the CAD/CAM Center of Dental Medicine, Research Institute, Department of Prosthetic Dental Medicine, Faculty of Dental Medicine, Medical University of Plovdiv, Bulgaria.

**Conflicts of Interest:** The authors declare no conflict of interest.

## Abbreviations

| | |
|---|---|
| AS | artificial saliva |
| AM | additive manufacturing |
| CAD/CAM | computer-aided design/computer-aided manufacturing |
| CD | complete dentures |
| ISO | International Organization of Standardization |
| PMMA | polymethyl methacrylate |
| STL | stereolithography, standard triangle language, standard tessellation language |
| 3D | three-dimensional |

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
