# Peer review of "Evaluation of Water Sorption and Solubility of 3D-Printed, CAD/CAM Milled, and PMMA Denture Base Materials Subjected to Artificial Aging"

_jcs, doi:10.3390/jcs7080339_

Round 1
Reviewer 1 Report (Previous Reviewer 1)
The authors did a good job. All my comments as a reviewer have been taken into account. As we wrote, the topic of 3D printing materials is very important in dentistry, because it is a new application of this technology. Thus, any work that investigates the effects of such materials, both in vitro and in vivo, is most welcome.T
Author Response
Dear Reviewer,
We would like to express our gratitude for your valuable feedback. Your comments and suggestions were very helpful for improving this manuscript. Insufficient research has been conducted on the characteristics of 3D-printed materials used for denture bases. We are pleased that our research will fill this gap and make a valuable contribution in this area. Thank you again for the support!
Best regards,
The authors
Reviewer 2 Report (New Reviewer)
Dear Authors,
thank you for this interesting article. Please, find the attached information how to improve it though:
1. Please, shorten the abstract to ca. 250 words (as suggested by Journal)
2. In the introduction you should write more about advantages of PMMA, especially when compared to polyamide-12 as a base denture material, eg.
- Wieckiewicz M, Opitz V, Richter G, Boening KW. Physical properties of polyamide-12 versus PMMA denture base material. Biomed Res Int. 2014;2014:150298. doi: 10.1155/2014/150298.
3. Also, in the introduction, you should add more information on the properties of 3D printed materials, such as stability in time, color stability etc, eg.:
- Kul E, Abdulrahim R, Bayındır F, Matori KA, Gül P. Evaluation of the color stability of temporary materials produced with CAD/CAM. Dent Med Probl. 2021;58(2):187–191. doi:10.17219/dmp/126745
- Grzebieluch, W.; Kowalewski, P.; Grygier, D.; Rutkowska-Gorczyca, M.; Kozakiewicz, M.; Jurczyszyn, K. Printable and Machinable Dental Restorative Composites for CAD/CAM Application—Comparison of Mechanical Properties, Fractographic, Texture and Fractal Dimension Analysis. Materials 2021, 14, 4919. https://doi.org/10.3390/ma14174919
Please, find more references
4. Lines 61-71, please complete the information on SM (one or two sentences) - this paragraph should focus on the definition with pointed out differences between AM and SM.
5. In the M&M section, please add the pictures of specimens used in the study (I would replace fig. 1 with the real picture). If possible, add the figures on how the experiment was performed (or a graph with the most important information, such as time, proportions, no of samples etc).
6. Line 171, please add the manufactureres' suggestions how to prepare the specimen
7. Line 180 - were both sides. polished?
8. In the discussion, please add the alternative methods for artificial aging, such as storage in water and thermocycling, eg.:
- Paradowska-Stolarz, A.; Wezgowiec, J.; Malysa, A.; Wieckiewicz, M. Effects of Polishing and Artificial Aging on Mechanical Properties of Dental LT Clear® Resin. J. Funct. Biomater. 2023, 14, 295. https://doi.org/10.3390/jfb14060295
- Duma, S.M.; Ilie, N. Adhesion to a CAD/CAM Composite: Causal Factors for a Reliable Long-Term Bond. J. Funct. Biomater. 2022, 13, 217. https://doi.org/10.3390/jfb13040217
It should be explained why you used this kind of artificial aging in the discussion, so that it leaves no doubts on the method to the reader.
9. Please, check the text in accordance to typos, eg. line 509 and more
10. Format the references (they are not in the same format).
Thank you
Author Response
Dear Reviewer,
Thank you very much for your valuable suggestions. Please see the specific answers to your comments below:
- Thank you, the abstract has been shortened to 250 words, according to the Journal’s instructions.
- The necessary information was added to the Introduction section.
- The Introduction section was revised, as you suggested. More references have been added.
- Thank you for your suggestion, the additional information has been added.
- We have revised and added a figure with the design of the study, we have removed figure 1.
- Thank you, additional details, according to the exact instructions of the manufacturer, have been added.
- Both sides of the test specimens were polished for the purposes of the conducted study.
- We have added the necessary information about the thermocycling and added the additional references.
- Thank you, the whole text will be carefully checked and revised.
- All the references have been formatted, according to the instructions.
Thank you kindly for the helpful comments! We are grateful for your valuable guidance.
Best regards,
The authors
Round 2
Reviewer 2 Report (New Reviewer)
Thank you for correcting the article according to the suggestions. In this form, the paper could be accepted.
This manuscript is a resubmission of an earlier submission. The following is a list of the peer review reports and author responses from that submission.
Round 1
Reviewer 1 Report
The work has been carefully corrected. There are still some changes, but it seems to me that they can be corrected while preparing the work for printing.
You have not yet described exactly how Thermocycling was carried out, in order to perform frequent temperature changes, you need a device and please specify its name and manufacturer.
In some rows, the degrees of Celcius from Superscript got to the normal line.
In the previous version, I didn't notice (I'm sorry, my fault) in figures 3,4,5 the sorption value and solubility on the Y axis are missing, you need to complete it.
Figure 6 - what do the colors green, yellow and pink mean?
Discussion
Gad et al.- is numbered in Ref [19].
All water sorption values of the tested 3D-printed specimens were lower than the ISO 464 recommendation for maximum water sorption [24]. After calculation, the average value 465 for water sorption (1.0) was found to be lower than the requirements of ISO.- these sentences are the same. Delete one.
Vallittu - it should be [25] - go through the numbering in the text again. Other publications are also shifted by 1 in relation to Ref.
line 502
bond density - what does it mean?
line 503
causing spatial heterogeneity - https://en.wikipedia.org/wiki/Spatial_heterogeneity - I would not use this term here, better heterogeneity in the polymerized material?
line 513
differences in solubility of MMA-based materials. PMMA and 3D printing results primarily from the degree of their cross-linking. PMMA has long chains, between which water can penetrate. 3D printing materials, like the composite cements you mention, are made of dimethacrylates or higher functional (3-6 methacylate or acrylate groups), so they are very densely cross-linked, which affects their lower sorption and solubility.
Thanks again for the work done. Just fix what I'm asking for and you'll be fine. Good luck.
OK
Reviewer 2 Report
The article still is not clearly presenting the results and conclusions.
There are no clinical implcations.
Reviewer 3 Report
Thank you for submitting your manuscript.
I had a look at the methodology then I found it not appropriate for a dental material research in terms for samples and preparations.
How did the authors calculate each specimen's volume (V) when they used a Square samples?
The authors stated that it's according to the to ISO standard (ISO 20795 1: 2013). However, the ISO required a disc-shape samples in order to calculate the specimen's volume NOT a Square.
Therefore, I recommended to reject the manuscript, and I strongly suggest that the authors should prepare new samples.
Thank
Extensive editing of English language required